# Organic Components of Small Bodies in the Outer Solar System: Some Results of the New Horizons Mission

**DOI:** 10.3390/life10080126

**Published:** 2020-07-28

**Authors:** Dale P. Cruikshank, Yvonne J. Pendleton, William M. Grundy

**Affiliations:** 1NASA Ames Research Center, Moffett Field, CA 94035, USA; yvonne.pendleton@nasa.gov; 2Lowell Observatory, Flagstaff, AZ 86001, USA; w.grundy@lowell.edu

**Keywords:** Solar System, Pluto, Charon, Arrokoth, Kuiper Belt object, complex organics, solar nebula, protoplanetary disk, New Horizons

## Abstract

The close encounters of the Pluto–Charon system and the Kuiper Belt object Arrokoth (formerly 2014 MU_69_) by NASA’s New Horizons spacecraft in 2015 and 2019, respectively, have given new perspectives on the most distant planetary bodies yet explored. These bodies are key indicators of the composition, chemistry, and dynamics of the outer regions of the Solar System’s nascent environment. Pluto and Charon reveal characteristics of the largest Kuiper Belt objects formed in the dynamically evolving solar nebula inward of ~30 AU, while the much smaller Arrokoth is a largely undisturbed relic of accretion at ~45 AU. The surfaces of Pluto and Charon are covered with volatile and refractory ices and organic components, and have been shaped by geological activity. On Pluto, N_2_, CO and CH_4_ are exchanged between the atmosphere and surface as gaseous and condensed phases on diurnal, seasonal and longer timescales, while Charon’s surface is primarily inert H_2_O ice with an ammoniated component and a polar region colored with a macromolecular organic deposit. Arrokoth is revealed as a fused binary body in a relatively benign space environment where it originated and has remained for the age of the Solar System. Its surface is a mix of CH_3_OH ice, a red-orange pigment of presumed complex organic material, and possibly other undetected components.

## 1. Introduction

The presence of complex organic molecules, often referred to as COMs, on outer Solar System planetary surfaces and in the atmosphere of Saturn’s satellite Titan were first suspected from their low albedos and reddish colors. Comparisons with terrestrial kerogens and complex refractory organics synthesized by photolysis and radiolysis of gaseous mixtures of CH_4_, N_2_, and other simple molecules have gradually led to the recognition that planetary processes in atmospheres and surface ices containing hydrocarbons lead to the formation of colored materials of relevance to both planetary environments and the interstellar medium, e.g., [1]. These results demonstrate that energetic processing of ices and gases containing simple organic molecules, such as CH_4_, readily transforms the initial carbon-bearing materials into large molecules with complex structures termed “tholin”. Complex organics formed in this way often have orange and red colors that closely resemble the distinctive colors commonly found on planetary bodies [2]. This material can be sufficiently non-volatile to be considered refractory, especially for analogs that are studied in room-temperature laboratory settings, but more volatile fractions that are only stable at low temperatures likely also play a role in cold, outer Solar System settings, e.g., [3]. Earth-based and spacecraft-based spectroscopy at visible and near-infrared wavelengths have confirmed the presence of both volatile and refractory organics on several planetary satellites, as well as Pluto and other large Kuiper Belt objects (abbreviated herein as KBOs). Gases and organic aerosols have been confirmed in the atmosphere of Pluto by UV spectroscopy [4], while mass spectroscopy with the Cassini spacecraft has found organics in the atmosphere of Titan [5,6] and the plumes of Enceladus [7]. In addition, refractory organic complexes have been found on the surfaces of Saturn’s satellites Phoebe, Iapetus, and Hyperion [8], Jupiter’s Ganymede and Callisto [9], Pluto and Charon [10,11], and the Kuiper Belt object (486958) 2014 MU_69_, now named Arrokoth [12,13]. Spectra and textures of surfaces consisting of ices, silicates, and tholins have been studied in the laboratory [14].

Here we present the current understanding of the organic constituents of these bodies with emphasis on KBOs, and in particular the results of the New Horizons investigations of the Pluto–Charon system and Arrokoth, and discuss the significance of the discoveries in terms of the origins of organics in the solar nebula and the protoplanetary disk.

The relevance of Pluto, Arrokoth, and other small bodies in the outer Solar System to investigations of the origin and evolution of the early Solar System is closely related to the distinct environments in which they formed in the protoplanetary nebula. These diverse origins are reflected today in the complex dynamical structure of the Kuiper belt [15]. Present day KBOs, the remnants of a much larger early planetesimal population, occupy a disk extending from the orbit of Neptune (~30 AU) to heliocentric distances of ~50 AU (astronomical units) and beyond. They are subdivided into classical, resonant, and scattered bodies. The classical KBOs are on relatively low inclination, low eccentricity orbits between 40 and 47 AU [16]. Their nearly circular orbits, and especially those of a sub-group of dynamically “cold” classical KBOs (CCKBOs), indicate that they have experienced little perturbation (i.e., low eccentricity and inclination) since they formed in the protoplanetary nebula [17,18,19,20]. Resonant objects occupy more dynamically excited orbits in mean-motion resonance with Neptune, with orbital periods that are in integer ratios with that of Neptune. Pluto is in the 2:3 mean-motion resonance and defines the subclass “plutinos”. Many other mean-motion resonances are well-populated, too, including 2:1, 3:1, 5:1, 5:2, 5:3, 7:3, and 7:4 [21], with a distribution that extends into even higher order resonances. Scattered objects also have high inclinations and eccentricities and a broad distribution of semimajor axes but are not in resonance with Neptune. The dynamical excitation of the resonant and scattered populations reveals much about the early evolution of the Solar System, and especially an outward migration of Neptune that cleared away most of the original planetesimal disk exterior to where Neptune formed. Neptune’s mean-motion resonances swept across the disk as Neptune migrated through it, resulting in capture of some planetesimals into the resonances [22,23,24,25,26]. The migration was driven by dynamical interaction between Neptune and the planetesimals, so the fact that Neptune stopped migrating at 30 AU indicates the existence of some sort of transition from a more massive, densely populated planetesimal disk inside 30 AU to a more depleted environment beyond that distance from the Sun [27].

Arrokoth is a cold classical KBO in a non-resonant orbit of low eccentricity (0.042) and low inclination (2.45°), with semi-major axis 44.58 AU [28]. These characteristics show that it accreted in the solar nebula in this near-circular orbit, and remained at its present distance for the age of the Solar System [12]. Thus Arrokoth samples the nebula beyond the 30-AU break that terminated Neptune’s migration, unlike Pluto, which, like other KBOs on more excited orbits, probably formed in the more densely populated region inside of 30 AU.

Another important class are the Centaur objects. These are an extension of the scattered population that have been recently (~10^7^ y) perturbed into orbits that are rapidly evolving through interaction with the giant planets [29,30]. Centaurs have relatively short dynamical lifetimes (~10^5^ to 10^8^ y) before being ejected from the Solar System or colliding with a planet. When perturbations to their orbits bring them closer to the Sun, the increased heating drives loss of mass through cometary activity [31]. A few have shown continuous cometary activity (e.g., comet 29P/Schwassmann-Wachmann 1, which is also regarded as a Centaur) or episodic activity (e.g., 2060 Chiron).

## 2. Kuiper Belt Object Arrokoth

The first in situ investigation of the nature and composition of a small Kuiper Belt object in its heliocentric region of origin was accomplished with the New Horizons spacecraft in 2019, when it flew by Arrokoth [12]. Dynamical calculations show that Arrokoth, with its near-circular and low inclination orbit, has not been substantially perturbed by the motion of the large planets over the age of the Solar System. The paucity of impact craters on its surface also points to a benign environment at that distance of ~45 AU. As such, Arrokoth provides an unusually valuable window on the environment and composition of the Solar System beyond the 30-AU edge of the main planetesimal disk.

Arrokoth is a fused binary body with combined length of 36 km and equivalent diameter of 18 km, and an estimated mean density of several hundred kg/m^3^ [32,33]. The surface of Arrokoth has a near-uniform orange-red color, and its near-infrared spectrum, obtained with the New Horizons LEISA instrument, shows absorption bands at 2.1, 2.27 and 2.33 μm identified as frozen methanol (CH_3_OH) (Figure 1). The presence of H_2_O ice on its surface is suggested by a shallow absorption band at 2.0 μm, but the band may be spurious, and the evidence for H_2_O is therefore ambiguous [13]. Water ice is expected on Arrokoth and all other KBOs, and has been detected spectroscopically on some, but not all, of the objects for which appropriate data exist. However, such data only exist for larger, brighter members of the more excited populations, not small cold classical KBOs like Arrokoth [34,35]. Both the steep slope of the reflectance spectrum of Arrokoth and the CH_3_OH bands are closely similar to those characteristics of Centaur 5145 Pholus, but Pholus also shows prominent H_2_O bands [36]. Another KBO, 2002VE_95_, a resonant plutino, also shows prominent CH_3_OH and H_2_O absorption bands [37], suggesting that the apparent absence of H_2_O on the surface of Arrokoth is anomalous. Whether or not ice is present on the surface, an opaque patina of space weathering is likely on these small bodies, and if present, may mask diagnostic spectral features of H_2_O and other possible components.

The dynamical stability of cold-classical KBOs over the age of the Solar System [38] demonstrates that the compositions of these bodies have not been altered by close approaches to the Sun. Furthermore, the small size of Arrokoth suggests that it has never been heated to a temperature at which it could melt and differentiate. Below the surface, the composition is expected to be a composite of simple ices, rocky chondritic material, and complex organic molecules inherited from the nascent interstellar cloud, and these may have been further processed in the protoplanetary disk [13,33,39]. We return to this point in Section 4.

## 3. Pluto and Charon

### 3.1. Pluto

With a radius of 1188.3 km, Pluto is about a hundred times larger than Arrokoth and ~10^7^ times more massive, and is representative of the largest Kuiper Belt objects. As previously described, like most other known large KBOs, also known as dwarf planets, it is likely to have accreted in the solar nebula inward of ~30 AU and was then forced to its present orbit by the outward migration of the giant planets [23,24]. Many of those bodies forced outward were captured into orbits resonant with Neptune, while others were scattered into orbits of high eccentricity and inclination. The formation of Pluto’s large satellite Charon (radius 606 km) is likely the result of a collision that left two of the bodies in locked, synchronous rotation after subsequent tidal evolution [40,41] and having similar, but not identical bulk densities (1854 kg/m^3^ for Pluto and 1702 kg/m^3^ for Charon) but different surface compositions [10].

Pluto’s variegated surface consists of a suite of highly volatile ices (N_2_, CO, CH_4_) that were originally found by near-infrared spectroscopy from ground-based telescopes [42] (Figure 2). These components were confirmed by infrared spectroscopy by the New Horizons spacecraft, and exposures of H_2_O ice and small amounts of frozen C_2_H_6_ and CH_3_OH were also found [10,43,44]. A non-volatile material on Pluto’s surface appears to consist of complex organic molecular material deposited from the atmosphere (see below) and produced directly on the surface by photolysis and radiolysis of the surface ices that contain the hydrocarbons CH_4_ and C_2_H_6_ [44]. Pluto’s variegated surface is shown in the color-enhanced image from New Horizons (Figure 3). Red-orange-colored H_2_O ice carries the spectral signature of NH_3_ hydrate, or possibly an ammoniated salt that is thought to have emerged from one or more subsurface reservoirs in episodes of cryovolcanism [45,46] (Figure 4). Beyond the surface expressions of H_2_O ice, there is no direct evidence of subsurface liquid water in Pluto’s interior, but the case has been made by various investigators for the existence of a global ocean at some time in the planet’s history, e.g., [47] on geodynamical considerations.

Grundy et al. [48] find that atmospheric aerosol tholin particles can result in thick surface deposits (several meters) over the history of Pluto. The range of colors seen on Pluto may result from chemical changes in the atmospheric tholins as they interact with the non-uniformly distributed volatiles on the planet’s surface, producing distinct products in different regions [49]. Alternatively, the supply of haze particles may vary with time, and possibly location, which would also produce different products in different places. A range of materials originating from the deposition of a macromolecular organic tholin onto volatile-rich surface regions may have prebiotic chemical significance.

Pluto is a dynamic planet in terms of current geological activity, as well as the exchange of volatiles between the surface and the atmosphere. It is also chemically dynamic as photolysis and radiolysis transform simple molecules into complex, macromolecular organic materials both in the atmosphere and on the surface. The colored H_2_O ice that has emerged onto the surface from internal reservoirs may represent another dimension of complex organic chemistry occurring in subsurface reservoirs, where liquid water interacts with the rocky components and various organic molecules incorporated from the solar nebula during the accretion of the planet [46]. It is uncertain whether the red color represents complex organics or if it originates in water–mineral interactions. Models of the evolution of fluids in the interiors of small bodies in the outer Solar System suggest that the interaction of water with rock assemblages and the principal components of comets result in the formation of hydrated silicates and antifreezes (e.g., NH_3_) that prolong the liquid state as the body cools [50]. The fluid is rich in reduced N and C, as well as dissolved H_2_, which is consistent with the presence of serpentines, ammonium phyllosilicates, and carbonates found on Ceres [51].

The formation of complex colored organic materials in fluids reacting with the organic components of chondritic meteorites [52] has been explored in the laboratory, especially in elucidating the role of formaldehyde polymer ((H_2_CO)n) in the formation of more complex material. Formaldehyde is found in interstellar ices and was likely incorporated into the solar nebula [53]. Kebukawa et al. [54,55] showed that in experiments simulating processes in aqueous environments in planetesimals, a suite of complex amino acids can be synthesized simultaneously with insoluble organic matter. Similarly, experiments with prolonged heating of solutions of formaldehyde, glycoaldehyde, and ammonia produced a dark refractory residue. This material of high molecular weight consisted of formaldehyde polymer in addition to olefinic and aromatic molecules, and was suggested to account for the dark reddish regions on Pluto [56].

Additional experiments show that the interaction of complex organics consisting of carbon-containing moieties in chains of combinations of functional groups in a hydrous environment, particularly in the presence of NH_3_, produces several amino acids [57,58]. Regardless of the degree of hydration, mixtures of amino acids and other organic complexes are readily produced in a variety of environments [59]. However, while fluid environments involving H_2_O have existed, or possibly still exist within Pluto and other KBOs sufficiently large to have melted to some extent, very small KBOs like Arrokoth are unlikely to have ever become warm enough inside to enable hydrolysis reactions, e.g., [13,60,61,62].

The significance to prebiotic chemistry of Pluto’s surface exposures of ammoniated H_2_O that contains a red pigment of presumed organic composition has been explored by Cruikshank et al. [63]. They note that laboratory experiments in which ultraviolet photolysis of water ice containing purines and pyrimidines (which have not been detected on Pluto or Arrokoth) and NH_3_ produces a range of nucleobases, including the five found in living systems on Earth.

### 3.2. Charon and Pluto’s Small Satellites

Images and near-infrared spectra of Charon obtained during the flyby of New Horizons in 2015 [10,45] have corroborated previous detections of NH_3_ or an ammoniated material on the surface found from ground-based near-infrared spectroscopy [64,65,66]. In addition, the spacecraft data have shown that the north polar region is tinted with an orange pigment [11] (Figure 5). The polar coloration can be explained by photolysis of CH_4_ deposited there during the alternating long-term darkness of each of Charon’s polar regions resulting from the high obliquity of the Pluto–Charon orbit. Methane gas is escaping from Pluto’s atmosphere [4], and a fraction of the streaming gas encounters Charon where it is cold-trapped on the polar region in darkness. That frozen CH_4_ is later photolyzed by solar Lyman-α radiation back-scattered by interplanetary hydrogen, producing a colored refractory residue (tholin). Unprocessed CH_4_ escapes Charon’s surface as direct sunlight is slowly restored during the orbital cycle, and the process is repeated for the south polar region (which is currently in darkness) over the 248-year orbital period of Pluto and Charon around the Sun [11].

The orange tholin on Charon’s north pole does not have diagnostic features in the region of the spectrum currently achievable from Earth-based data. In addition to ground-based spectra extending to 2.5 μm obtained by Brown and Calvin [64] and other investigators, Protopapa et al. [67] recorded the spectrum to ~4.05 μm, but at a low signal precision that did not reveal identifiable absorption bands. Although the New Horizons spacecraft obtained high-quality, spatially resolved spectral images of Charon (1.1–2.5 μm), no new diagnostic bands were found [10]. Dalle Ore et al. [68] used the New Horizons data to map spatial variability in the distribution of tholin and ices across the surface, finding differences in the abundances of crystalline and amorphous H_2_O ice. Overall, the spectrum of Charon is well modeled by a combination of crystalline and amorphous H_2_O ice, an ammonia hydrate (NH_3_·H_2_O), plus a tholin made from the electron radiolysis of a mixture of CH_4_, N_2_, and CO ices [68].

In the New Horizons data, at least two of Pluto’s very small satellites, Nix and Hydra, also show the presence of H_2_O ice, an ammonia compound that is probably a hydrate but possibly an ammonium salt, and, in the case of Nix, a distinct reddish color on one portion of the surface [69]. Whether or not the reddish color represents a deposit of complex organics is not known.

An ammonia spectral signature is also found on some exposures of H_2_O ice on Pluto [45,46], but as in the case of Charon and the small satellites, the identity of the ammoniated species is ambiguous. Pure ammonia ice on an exposed surface is not stable against UV irradiation, and although the stability of an ammonia hydrate is greater, it too is expected to disappear on relatively short timescales. On the basis of laboratory experiments with proton-irradiated NH_3_·H_2_O, Loeffler et al. [70] estimate that over the age of the Solar System, about 40% or more of any original ammonia on Charon has been removed from the optical surface. The stability of ammoniated salts is expected to be greater, but quantitative estimates await appropriate laboratory experiments.

Another obstacle to a complete understanding of the surface chemistries of Charon and the small satellites is that tholins are expected to lose their color as the complex molecules are broken down by prolonged photolysis or radiolysis, eventually being converted to carbon, either in an amorphous state or as graphite. A limited laboratory study of the effect of long-term irradiation of a methane clathrate showed that as the accumulated dose of charged particles increased, the originally high-albedo colorless solid developed a distinct red color, which was eventually quenched as the albedo dropped to ~10 to 20% of its original value [71]. The overall neutral gray color of most of Charon’s surface (apart from the reddish north polar region) may be explained by the long-term exposure of a veneer of organics on its surface. This situation does not apply to Pluto, where the surface is continually altered by the deposition of atmospheric aerosols in addition to the exchange of volatiles between the atmosphere and surface on seasonal and long-term timescales. The apparent indefinite retention of a red-colored region on Nix is not explained, but may hint at more recent delivery via an impact.

## 4. The Formation of Arrokoth in the Solar Nebula and Contrasts to Pluto

The detection of frozen methanol and a nearly uniform red-orange coloration across both lobes of the small body of Arrokoth are clues to its origin and evolution. The relatively benign environment in which Arrokoth formed is indicated not only by its low inclination and nearly circular orbit at ~45 AU, but also by the paucity of large craters on both lobes of the body. Dynamical arguments and population studies of the cold-classical KBOs provide a convincing case for the separate formation of both lobes prior to their joining in a very gentle collision (~3 m/s) that occurred early in the formation of the Solar System [33]. The details of accretion are therefore likely to be well preserved within Arrokoth, reflected in the stability of ices found at the present time [72]. Pluto, having formed from material significantly closer to the Sun, was later perturbed outward to its current location by the migration of Neptune [23,24], providing an important contrast to the formation of Arrokoth. Pluto’s much larger size (factor of about one hundred in linear dimensions and two million in volume) and tenuous atmosphere make its present-day composition decidedly different from that of Arrokoth, but comparing them can provide a context in which to place the compositions of other Kuiper Belt Objects with more complexity than Arrokoth and possibly less than Pluto.

Although CH_3_OH is the only ice component detected on the present-day surface of Arrokoth, its presence in the interstellar medium (ISM), e.g., [73] and in comets, e.g., [74] invites a deeper comparison. We note that CH_3_OH is not highly volatile and is more like H_2_O ice in its stability on planetary bodies. In the absence of any other plausible cause, and in view of numerous synthesis studies of tholins in the laboratory, the red-orange coloration of the surface is regarded as strongly suggestive of complex macromolecular organics found or inferred on several outer Solar System bodies [2]. Other volatile ices may exist on the surface of Arrokoth but at concentrations below the detection threshold of the LEISA spectrometer on the New Horizons spacecraft. Candidates are N_2_, CO, and CH_4_. While these volatiles could be transiently stable on Arrokoth’s coldest winter surface regions, none can survive summer surface temperatures, which range from 40 to 60 K, nor could they survive in Arrokoth’s interior, where the mean temperature is ~40 K [13]. However, some could persist trapped in clathrate form or within amorphous ice. If these species were originally present in the interior, they would have been lost from the comet’s outer layers through outgassing that may have produced what appear to be collapse pits on the body’s surface [13]. Pits and sinkholes are also found on comet 67P/Churumov–Gerasimenko [75,76].

The color of Arrokoth is nearly uniform, with the major exception being a lighter tone in the neck where the two bodies are fused. On many Solar System bodies, a colored surface is thought to represent a relatively thin veneer of space-weathered material. While there are no fresh craters or fault surfaces that reveal the subsurface, it is plausible that the color runs deeper than the optical surface. Considerations of surface erosion from images with resolution up to ~33 m/pixel suggest that, over the age of Arrokoth, as much as a few meters of the surface have been removed in an ongoing process by impacts of micrometeorites from sources in the Solar System and interstellar space [77,78]. If this process removes material faster than photolysis and radiolysis can produce the color from native material, the color may be representative of the bulk composition of both components of Arrokoth [13]. Or, if the timescale for photolysis of native hydrocarbons is comparable to the erosion and exposure timescale, the current color of the surface represents a snapshot in time of long-term and ongoing processes (any present-day feed stocks would have to be hydrocarbons that are less volatile than methane or else be trapped in clathrate or amorphous ice, since methane ice itself is too volatile, e.g., [13,72,79]). If, in fact, Arrokoth accreted in the solar nebula before the disk had cleared sufficiently to expose its native feedstock to UV photolysis by the Sun, it would seem that the original material in the protoplanetary disk had already been processed sufficiently to produce the complex refractory tholins that represent the red-orange material.

## 5. Forming Planetary Systems

The proto-planetary disk (PPD) phase, which occurs prior to the development of solar systems, can be effectively modeled through a combination of the expected physical and chemical processes plus constraints offered from observational data, e.g., [80]. Dynamical movement of the gas and dust, the physical conditions of the solar nebula, and the composition of the PPD drive the chemical evolution at every stage. Chemical reactions occur as the ice-mantled dust grains cycle vertically and radially through the PPD, e.g., [81]. Observations that can spatially resolve details in some of the nearby forming systems are critical to provide insight at different stages along the way, e.g., [82]. 

For instance, recent observations have revealed a depletion of gas phase CO in the warm (T > 22 K) upper regions of young PPDs up to two orders of magnitude below the expected canonical value of 10^−4^, e.g., [83,84], meaning that the solid state CO on the dust grains did not go back into the gas phase as thoroughly as expected. To explain the absence of the CO in the gas phase, investigators have suggested more CO than previously thought could have been sequestered on icy grains shielded in the mid-plane of the disk, e.g., [85] or that the CO may participate more fully in additional solid state chemistry that occurs on the grains through the addition of hydrogen to form products such as HCO, H_2_CO, methanol, etc. [86,87]. Indeed, the PPD models that use CO to form other molecules can produce methanol on the relevant timescales (~Myr) required if the lobes of Arrokoth incorporated methanol from the planetesimal stage. However, much work is yet to be done to understand the distribution over time and distance within the PPD that will result when the strictly physical sequestration approach is combined with the chemical evolution of CO. Arrokoth presents an intriguing data point because it formed where we see it today, both components are thought to have fully formed as planetesimals early on, and it has existed in a benign environment over the age of the Solar System. It will be interesting to see if the future PPD models yield results that can explain the New Horizons observations of methanol on Arrokoth, not only on the required timescale, but also at the distance where it exists from our Sun. Similarly, it will be interesting to see what the future models predict for water and volatiles that were not detected (on the surface) of Arrokoth. The coupling of these sophisticated modeling efforts with future observations of distant star and planetary systems in formation, plus forthcoming new data from primitive objects in our own Solar System, will greatly advance our understanding of the origin and evolution of ices and organics in planetary systems.

## Figures and Tables

**Figure 1 life-10-00126-f001:**
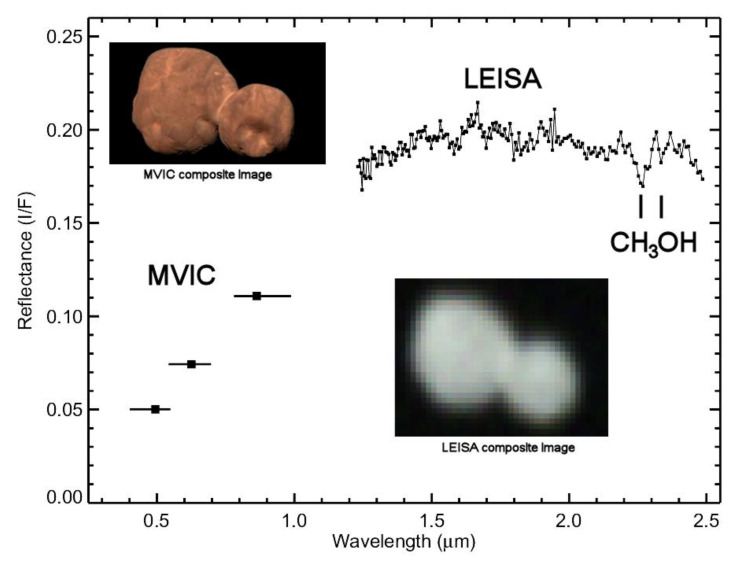
The spectrum and images of Arrokoth obtained with the New Horizons spacecraft. The color image was compiled from images through three color filters with the MVIC (Multispectral Visible Imaging Camera). The spectrum from 1.2 to 2.5 μm was derived from spectral images with LEISA (Linear Etalon Imaging Spectral Array). The positions of the two absorption bands attributed to frozen methanol (CH_3_OH) are marked. The LEISA image is an rgb composite from two wavelengths, 1.6 and 2.0 μm.

**Figure 2 life-10-00126-f002:**
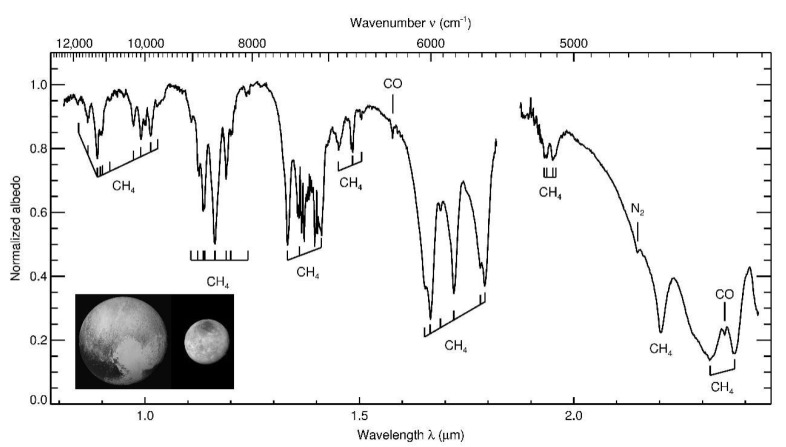
Infrared spectrum of Pluto (including the negligible light from Charon), shown in the inset, compiled from 20 years of ground-based telescopic observations. The identified absorption bands of CH_4_, N_2_, and CO ices are marked. Data from B. J. Holler et al. (in preparation).

**Figure 3 life-10-00126-f003:**
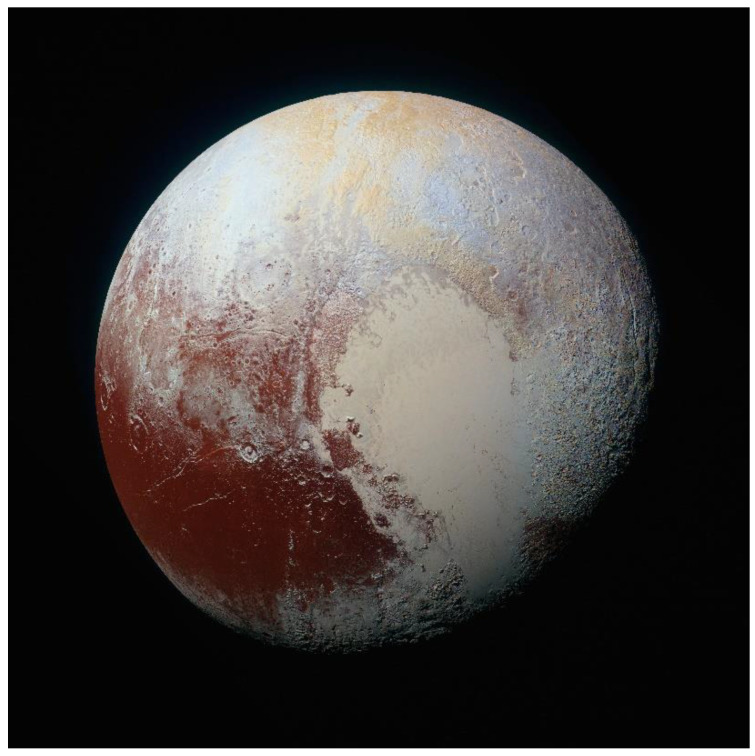
Color-enhanced full-disk view of the encounter hemisphere of Pluto from the New Horizons spacecraft. NASA image, courtesy of Southwest Research Institute and Johns Hopkins University.

**Figure 4 life-10-00126-f004:**
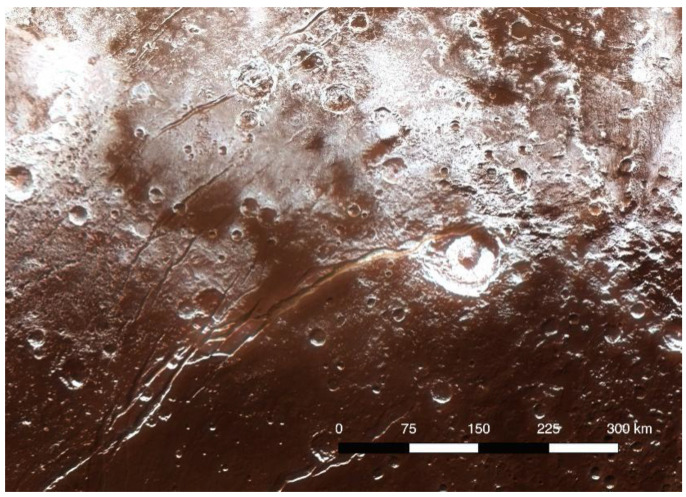
The region around Elliot crater (center right) and the Virgil Fossae complex, showing the strong red coloration in the main fossa trough and surrounding terrain. The red color corresponds to H_2_O ice that is lightly laced with an ammonia compound. NASA image, courtesy of Southwest Research Institute and Johns Hopkins University.

**Figure 5 life-10-00126-f005:**
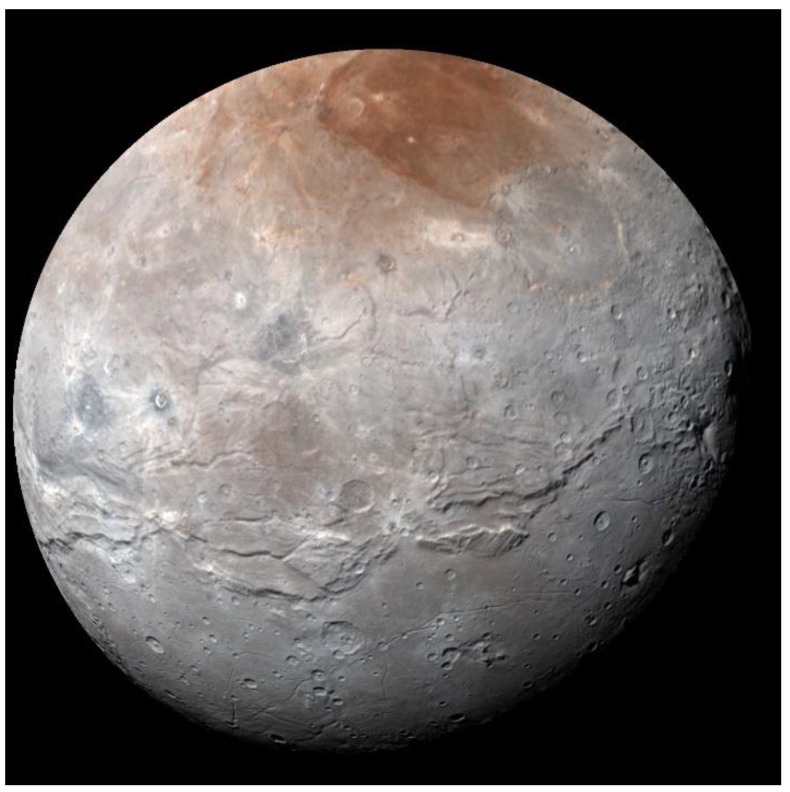
The Pluto-facing hemisphere of Charon, imaged with the New Horizons spacecraft, showing overall gray color with the orange tinted north polar region. NASA image, courtesy of Southwest Research Institute and Johns Hopkins University.

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
