# Peer review of "Organic Components of Small Bodies in the Outer Solar System: Some Results of the New Horizons Mission"

_life, 2020, doi:10.3390/life10080126_

Round 1
Reviewer 1 Report
In general, using the acronym KBO helps the writing but does not fluidify the reading. I suggest to use the full name.
Line 262: "Furthermore, the red-orange coloration of the surface is indicative of complex organic molecules similar to tholins synthesized in 263 the laboratory and found on several outer Solar System bodies."
The statement is too strong. Fior a chemist, the coloration is not sufficient to "indicate" organic molecules. More recent references about tholins should be added.
Author Response
Review of "Life" paper. Revisions made by authors in response to reviewers. 7.12.2020
Reviewer 1 Comments and Suggestions for Authors
In general, using the acronym KBO helps the writing but does not fluidify the reading. I suggest to use the full name.
RESPONSE: On line 60, we note that Kuiper Belt objects are abbreviated herein as KBOs), and this is avoids spelling it out each time the term is used.
Line 262: "Furthermore, the red-orange coloration of the surface is indicative of complex organic molecules similar to tholins synthesized in the laboratory and found on several outer Solar System bodies."
The statement is too strong. For a chemist, the coloration is not sufficient to "indicate" organic molecules. More recent references about tholins should be added.
RESPONSE: This statement now reads:
"In the absence of any other plausible cause, and in view of numerous synthesis studies of tholins in the laboratory, the red-orange coloration of the surface is regarded as strongly suggestive of complex macromolecular organics found or inferred on several outer Solar System bodies (Cruikshank et al. 2005)."
Two recent references to tholins are added (Brasse et al. and Poch et al.).
Brassé, C; Muñoz, O.; Coll, P.; Rauln, F. Optical constants of Titan aerosols and there tholins analogs: Experimental results and modeling/observational data. Planet. Space. Sci. 2015, 100-110, pp. 159-174.
Poch, O.; Pommerol, A.; Jost, B. et al. 2016. Sublimation of water ice mixed with silicates and tholins: Evolution of surface texture and reflectance spectra, with implications for comets. Icarus 267 (2018) 154-173.
Reviewer 2 Comments and Suggestions for Authors
This is a nice, albeit quite brief, summary of the findings from New Horizons. However, the authors should elaborate on a few points further before it merits publication in the journal.
(1) One point that struck me immediately while reading this manuscript is the complete absence of figures. This is strange for an observation-centric work, and makes it difficult to visualize many of the findings stated by the authors. Hence, they should either endeavour to include some data or at least schematic figures.
RESPONSE: Good idea. We have added some figures to the paper to augment the text.
(2) In the discussion of different ices on the surface of Pluto (and Charon), I would like to see more details about how these conclusions were obtained. For instance, was it through spectrometry, comparison with lab experiments, etc. It is important to provide these details to make the work self-contained.
RESPONSE: The relevant paragraph has been modified in response to the reviewer's request:
"Pluto's variegated surface consists of a suite of highly volatile ices (N2, CO, CH4) that were originally found by near-infrared spectroscopy from ground-based telescopes (see review by Cruikshank et al. 2015). These components were confirmed by infrared spectroscopy by the New Horizons spacecraft, and exposures of H2O ice and small amounts of frozen C2H6 and CH3OH were also found (Stern et al. 2015; Grundy et al. 2016a; Cook et al. 2019)."
For Charon, the relevant sentences have been modified thusly:
"Images and near-infrared spectra of Charon obtained during the flyby of New Horizons in 2015 (Stern et al. 2016; Grundy et al. 2016a) have corroborated previous detections of NH3 or an ammoniated material on the surface found from ground-based near-infrared spectroscopy (e.g., Brown and Calvin 2000; Dumas et al. 2001; Cook et al. 2007). In addition, the spacecraft data have shown that the north polar region is tinted with an orange pigment (Grundy et al. 2016b)."
RESPONSE: Additional explanation is now given.
(3) Did New Horizons provide any (in)direct evidence for subsurface water ocean on Pluto? If so, this is worth pointing out.
RESPONSE: This sentence has been added in the appropriate place:
"Beyond the surface expressions of H2O ice, there is no direct evidence of subsurface liquid water in Pluto's interior, but the case has been made by various investigators for the existence of a global ocean at some time in the planet's history (e.g., Nimmo et al. 2016; McKinnon et al.) on geodynamical considerations."
(4) The Cleaves et al. (2014) reference on line 185 was specifically for Titan-like tholins in contact with liquid ammonia. I don't think this context applies to Pluto; hence a suitable caveat should be added, or the reference removed.
RESPONSE: The phrase "in a variety of environments" preceding the Cleaves reference can be broadly interpreted to include an ammonia-rich environment. The reference is retained.
(5) In the discussion on lines 241-244, it is explained how Charon may have come about its gray colour: through the degradation of organics. The authors should briefly compare and contrast with Pluto, and explain the differences here.
RESPONSE: The following sentence has been added in the appropriate place:
"This situation does not apply to Pluto, where the surface is continually altered by the deposition of atmospheric aerosols in addition to the exchange of volatiles between the atmosphere and surface on seasonal and long-term timescales.
Reviewer 2 Report
This is a nice, albeit quite brief, summary of the findings from New Horizons. However, the authors should elaborate on a few points further before it merits publication in the journal.
(1) One point that struck me immediately while reading this manuscript is the complete absence of figures. This is strange for an observation-centric work, and makes it difficult to visualize many of the findings stated by the authors. Hence, they should either endeavour to include some data or at least schematic figures.
(2) In the discussion of different ices on the surface of Pluto (and Charon), I would like to see more details about how these conclusions were obtained. For instance, was it through spectrometry, comparison with lab experiments, etc. It is important to provide these details to make the work self-contained.
(3) Did New Horizons provide any (in)direct evidence for subsurface water ocean on Pluto? If so, this is worth pointing out.
(4) The Cleaves et al. (2014) reference on line 185 was specifically for Titan-like tholins in contact with liquid ammonia. I don't think this context applies to Pluto; hence a suitable caveat should be added, or the reference removed.
(5) In the discussion on lines 241-244, it is explained hot Charon may have come about its gray colour: through the degradation of organics. The authors should briefly compare and contrast with Pluto, and explain the differences here.
Author Response

(The authors gave the same response as above.)
